# Increased Interleukin-11 and Stress-Related Gene Expression in Human Endothelial and Bronchial Epithelial Cells Exposed to Silver Nanoparticles

**DOI:** 10.3390/biom11020234

**Published:** 2021-02-07

**Authors:** Jiyoung Jang, Sun Park, In-Hong Choi

**Affiliations:** 1Department of Microbiology, Yonsei University College of Medicine, Seoul 03722, Korea; Jang191111@korea.kr; 2Institute for Immunology and Immunological Diseases, Yonsei University College of Medicine, Seoul 03722, Korea; 3Humidifier Disinfectant Health Center, National Institute of Environmental Research, Incheon 22689, Korea; 4Department of Biomedical Sciences, The Graduate School, Ajou University, Suwon 16499, Korea; 5Department of Microbiology, Ajou University School of Medicine, Suwon 16499, Korea

**Keywords:** silver nanoparticle, interleukin-11, oxidative stress, microarray

## Abstract

This article aimed to identify and distinguish the various responses to silver nanoparticles (NPs) of endothelial and epithelial cells. We also assessed the significantly increased gene expression levels, as shown by microarray analysis. We evaluated the median lethal dose of NPs in each cell line and found that each value was different. We also confirmed the toxicity of 5 nm silver NPs. Meanwhile, cell death was not observed in cells exposed to 100 nm silver NPs at a high concentration. We verified that 5 nm silver NPs affected the variation in gene expression in cells through microarray analysis and observed a noticeable increase in interleukin (IL)-8 and IL-11 gene expression in early stages. This study showed noticeable variation in the expression of oxidative stress-related genes in early stages. Microarray results showed considerable variation in cell death-, apoptosis-, and cell survival-related gene expression. Of note, IL-11 gene expression was particularly increased following the exposure of endothelial and epithelial cells to 5 nm silver NPs. In conclusion, this study demonstrated that intracellular genes specifically responded to silver NPs in respiratory epithelial cells and endothelial cells. Among cytokine genes, IL-11 expression was noticeably increased. Additionally, we confirmed that NP toxicity was affected by NP size and dose.

## 1. Introduction

The ultrafine particle size of nanomaterials is limited to approximately 100 nm, and nanomaterials have been applied in various ways due to this small size [1]. Currently, nanotechnology is one of the leading scientific fields, combining the fields of physics, chemistry, biology, medicine, pharmaceutical science, informatics, and engineering [2]. Nanotechnology represents an emerging dynamic field with approximately 50,000 articles being published each year, and according to the European Patent Office, >2500 patents have been filed recently [3].

Particularly, metal nanoparticles (NPs) that contain gold, silver, iron, zinc, and metal oxides have been widely used owing to their large surface-area-to-volume ratio and unique physicochemical properties, including high electrical and thermal conductivity and optical, magnetic, and catalytic activities [4,5]. However, silver NPs have been identified as a cause of toxicity in the human body; therefore, they are considered dual-natured with both positive and negative aspects. Consequently, many toxicity evaluations and risk assessments of NPs are being conducted.

The endothelium is the human body’s first internal layer of blood vessels and is distributed throughout the body. Therefore, the endothelium allows molecules to move to and circulate between various tissues through the blood. Moreover, the endothelium controls certain pathways, such as lipid metabolism and vascular inflammation. When these functions are lost, vascular diseases may be triggered [6]. These characteristics of the endothelium are closely linked to the mechanism by which NPs infiltrate the human body and enter various organs.

In this study, we evaluated the gene expression of endothelial cells and epithelial cells when exposed to silver NPs. Among cytokine genes, the expression of interleukin (IL-11) as well as IL-8 were noticeable. The increased IL-8 levels due to silver NPs has been reported by us previously [7]; therefore, we focused on IL-11 in the present study. IL-11 is a member of the IL-6 family of cytokines and performs various functions in various cells. The IL-6 family of cytokines includes not only IL-11, but also IL-6, IL-27, IL-31, leukemia inhibitory factor, ciliary neurotrophic factor, oncostatin M, and cardiotrophin-1 [8]. IL-6 is associated with chronic inflammatory diseases and is responsible for the progression of various types of cancer. It activates the signal transducer and the activator of transcription 3 (STAT3) pro-survival pathways. According to one study, IL-6 and IL-11 were responsible for advancing gastrointestinal cancer, and during this process, IL-11 was found to be closely associated with STAT3 activation [9]. IL-11 performs hematopoietic functions in many different areas of the body, including the liver, gastrointestinal tract, lungs, heart, central nervous system, bones, joints, and immune system [10]. Playing a protective role by accelerating platelet recovery and reducing inflammatory responses, IL-11 was able to lower the mortality rate among sepsis patients [11]. In other words, among the immune responses of IL-11, anti-inflammatory properties have also been identified. Consequently, the various functions and roles of IL-11 are drawing new attention. Other biological activities of IL-11 include the stimulation of erythropoiesis and activation of megakaryocytes. Moreover, it controls the polarization of T cells and macrophages and promotes the maturation of bone-resorbing osteoclasts [12]. It also plays a role in neurogenesis [13], adipogenesis [14], and the promotion of stem cell development [15]. Additionally, it protects cells from graft versus host disease [16] and blocks gastric acid secretion [17]. Finally, IL-11 performs a variety of functions in various cells. IL-11 reduces proliferation and induces apoptosis of epithelial cells. In endothelial cells, it plays a role in cell activation and expression of surviving cell proliferation-related proteins [18]. IL-11 also affects mast cells, inducing their proliferation. Moreover, in macrophages and osteoclasts, respectively, IL-11 reduces the production of IL-1β, IL-12, NO, and NF-κB and decreases formation, increasing bone resorption [17].

In this study, we assessed IL-11 production upon exposure to silver NPs in endothelial cells and respiratory epithelial cells, because IL-11 has various important functions as described above, and the roles can vary by cell types [18]. Therefore, our results will provide important insights into better understanding the effects of IL-11 produced upon exposure silver NPs.

## 2. Materials and Methods

### 2.1. Silver NPs

Silver NPs in water-based solutions were obtained from Dr. Koh (5 nm; PNM, Hwaseong, Korea) or purchased from ABC Nanotech (100 nm; Daejeon, Korea). All silver NPs were round and PVP-coated. For cell culture, silver NPs at different concentrations were prepared in high-glucose Dulbecco’s Modified Eagle’s Medium (DMEM; Welgenen, Gyeongsan, Korea) and 2-mM L-glutamine-containing Minimum Essential Medium (MEM; Welgenen) supplemented with 10% fetal bovine serum (FBS; Welgenen), penicillin (100 U/mL), and streptomycin (100 µg/mL).

### 2.2. Characterization of Silver NPs

The diameter of the silver NPs was determined using transmission electron microscopy (TEM; model JEM-2010, JEOL Ltd., Tokyo, Japan). Agglomeration of NPs (0.5 mg/mL) in DMEM with 10% FBS was examined by dynamic light scattering (DLS; Malvern Instruments, Novato, CA, USA).

### 2.3. Cell Lines and Culture

EA.hy926 cells (human umbilical vein cell line; ATCC, Manassas, VA, USA) were cultured in DMEM, and BEAS-2B cells (human bronchial epithelial cell line; ATCC) were cultured in MEM containing 10% FBS and penicillin–streptomycin (100 U/mL and 100 µg/mL, respectively) at 37 °C in a humidified 5% CO_2_ atmosphere in an incubator. Although endotoxins were not detected in the silver NPs used in this study, polymyxin B (10 ng/mL; InvivoGen, San Diego, CA, USA) was added as an endotoxin neutralizer.

### 2.4. Analysis of Cell Proliferation

Cell viability was assessed using the CCK-8 Kit (Dojindo Laboratories, Kyoto, Japan). EA.hy926 (5 × 10^4^ cells/well) and BEAS-2B (6 × 10^4^ cells/well) cells were plated in 24-well plates and incubated overnight at 37 °C in a 5% CO_2_ incubator. The medium was removed, and then EA.hy926 cells were treated with silver NP solution (500 µL) diluted in the growth medium, whereas BEAS-2B cells were treated with silver NP solution (500 µL) diluted in 2% FBS-containing growth medium. After 24 h, CCK-8 reagent (15 µL) was added to each well, followed by incubation at 37 °C for 2 h. After centrifugation at 13,000 rpm for 5 min, EA.hy926 (200 µL) and BEAS-2B (100 µL) supernatants were transferred to 96-well microtiter plates, and optical density was measured at 450 nm with a spectrophotometer (BioTek Instruments, Winooski, VT, USA) to ensure that no optical interference was induced by the silver NPs.

### 2.5. Cytokine Detection

Enzyme-linked immunosorbent assay (ELISA) was performed to detect the presence of cytokines IL-8 and IL-11. EA.hy926 cells (1.5 × 10^5^ cells/well) were plated in DMEM (2 mL) containing 10% FBS in 6-well plates. BEAS-2B cells (2 × 10^5^ cells/well) were plated in MEM (2 mL) containing 10% FBS in 6-well plates. The medium was then removed, and silver NPs suspended in the growth medium and silver NPs suspended in 2% FBS-containing growth medium were added to each well containing EA.hy926 and BEAS-2B cells, respectively, to obtain a final volume of 1 mL per well. ELISA was performed using Human Cytokine IL-8 (BD Biosciences, San Diego, CA, USA) and IL-11 (R&D Systems, Minneapolis, MN, USA) Assay Kits. These kits use biotinylated anti-IL-8 or anti-IL-11 antibodies and streptavidin conjugated to horseradish–peroxidase. The OD was measured at 450 nm.

### 2.6. Real-Time Reverse Transcription Polymerase Chain Reaction (RT-PCR)

cDNA was synthesized from total RNA via reverse transcription with random primers (Invitrogen, San Diego, CA, USA). Primer pairs designed to amplify the cDNA encoding the target genes were prepared using the Invitrogen Oligo Perfect Designer (Thermo Fisher Scientific, Waltham, MA, USA). PCR reactions were performed using FastStart Universal SYBR Green Master (ROX) reagent according to the manufacturer’s instructions (Roche Applied Science, Mannheim, Germany) in a 7500 and StepOne Plus Real-Time PCR system (Applied Biosystems, Foster City, CA, USA). The reaction parameters were as follows: 2 min at 50 °C, 10 min at 95 °C, 40 cycles of denaturation at 95 °C for 15 s, and 60 °C for 1 min. Real-time RT-PCR data for each gene product were normalized against levels of glyceraldehyde 3-phosphate dehydrogenase (GAPDH). All transcript levels were reported as mean ± standard deviation (SD) relative to untreated controls from triplicate analyses. Gene expression levels were analyzed using the comparative CT method with the fold difference calculated based on the endogenous control (GAPDH) [19]. Primer sequences were as follows: IL-8, forward: 5′-GTGCAGTTTTGCCAAGGAGT-3′ and reverse: 5′-CTCTGCACCCAGTTTTCCTT-3′; IL-11, forward: 5′-CTGAGCCTGTGGCCAGATA-3′ and reverse: 5′-AGCTGTAGAGCTCCCAGTGC-3′; hemeoxygenase-1 (HO-1), forward: 5′-ATGACACCAAGGACCAGAGC-3′ and reverse: 5′-GTGTAAGGACCCATCGGAGA-3′; heat shock protein 70 kDa (HSP-70), forward: 5′-AGGCCAACAAGATCACCA-3′ and reverse: 5′-TCGTCCTCCGCTTTGTACTT-3′; and glyceraldehyde-3-phosphate dehydrogenase (GAPDH), forward: 5′-GATCATCAATGCCTCCT-3′ and reverse: 5′-TGTGGTCATGAGTCCTTCCA-3′.

### 2.7. Western Blotting

Cells were treated with 5 nm and 100 nm silver NPs for 24 h. Cells were harvested and lysed using lysis buffer (200 µL; 150 mM NaCl, 1% NP-40, 0.1% sodium dodecyl sulfate (SDS), 50 mM Tris (pH 8), 5 mM NaF, 1 mM Na_3_VO_4_, 1 mM PMSF, protease inhibitor cocktail) at 4 °C for 2 h. Cell lysates were centrifuged at 13,000 rpm at 4 °C for 15 min, and the supernatants were stored at −20 °C. Protein (30–40 µg) from each sample was boiled for 5 min and loaded on a 12% SDS–polyacrylamide gel. After electrophoresis, proteins in the gel were transferred onto nitrocellulose membranes (Amersham, Glattbrugg, Switzerland). After blocking with 5% skim milk (BD Biosciences, Franklin, NJ, USA), membranes were reacted overnight with primary antibodies, including anti-HO-1 antibodies (Cell Signaling Technology, Danvers, MA, USA) and anti-HSP-70 antibodies (Cell Signaling Technology), at 1:1000 dilution in 5% bovine serum albumin (Affymetrix, Cleveland, OH, USA) at 4 °C. After washing, the membranes were further reacted with a secondary antibody (peroxidase-conjugated AffiniPure goat anti-rabbit IgG; Jackson ImmunoResearch, West Grove, PA, USA) at 1:2000 dilution in 5% skim milk at room temperature for 1 h. Anti-GAPDH antibody (Cell Signaling Technology) was used to assess the transcription levels of housekeeping genes.

### 2.8. TEM Analysis

TEM analysis was performed according to the procedure described by DaeHyoun et al., (2012) [18].

### 2.9. RNA Isolation and cDNA Microarray Analysis

Total RNA was extracted using the RNeasy^®^ Mini Kit (Qiagen, Hilden, Germany), according to the manufacturer’s guidelines. RNA quantity was measured using a Nanodrop ND-1000 spectrophotometer (Wilmington, DE, USA), and an RNA 260/280 ratio of 1.8–2.1 was applied to all samples. Whole genome microarray analysis was performed using an AffymetrixGeneChip^®^ Human Gene 2.0 ST Array (Affymetrix, Santa Clara, CA, USA). EA.hy926 cells (4.5 × 10^5^ cells) were plated on 60-mm Petri dishes overnight and then exposed to 5 nm or 100 nm silver NPs (1.5 and 2 µg/mL) for 6 h. BEAS-2B cells (3.5 × 10^5^ cells) were plated on 60-mm Petri dishes overnight and then exposed to 5 nm or 100 nm silver NPs (0.25, 0.5 and 0.75 µg/mL) for 6 h. cDNA was synthesized from total RNA (3 µg) in the presence of a random primer, and cDNA microarray results were analyzed using independent *t*-tests.

### 2.10. Statistical Analysis

One-way analysis of variance (ANOVA) and two-way ANOVA with Bonferroni post-tests were performed for statistical comparisons for two groups and more than two groups, respectively. For cDNA microarray analysis, gene expression values were median-centered and imported into the Expression Console 1.4 software. Principal component analysis was performed to verify the consistency of the experiments and determine the presence of any chip outliers. Transcripts with a 1.5-fold (EA.hy926) and 2-fold (BEAS-2B) or higher change in their expression values were selected, and a t-test was performed to evaluate the significance of the differences. *p*-values < 0.05 were considered statistically significant.

## 3. Results

### 3.1. Characterization of Silver NPs

Characterization of silver NPs used in this experiment is shown in Figure 1. The 5 nm silver NPs were comparatively consistent in size, whereas the 100 nm silver NPs varied in size. Particle shape was round (Figure 1A). DLS analysis showed that the hydrodynamic diameters of the 5 nm and 100 nm silver NPs were 5.3 nm and 104.7 nm, respectively (Figure 1B).

### 3.2. Cytotoxicity in Endothelial and Epithelial Cells

As shown in Figure 2, cell viability was decreased with the increase in the concentration of 5 nm silver NPs. At silver NP concentrations of 1, 1.5, 2, 2.5, 2.75, 3, and 3.5 µg/mL, cell viability of EA.hy926 cells was reduced to 91%, 69%, 26%, 8%, 4%, 3%, and 1%, respectively; viability was significantly lower in the silver NP-exposed cells than in the control cells. The median lethal dose (LD_50_) of 5 nm silver NPs was approximately 1.72 µg/mL. However, 100 nm silver NPs at up to 3.5 µg/mL did not show cytotoxicity (Figure 2A). At silver NP concentrations of 0.5, 0.75, 1, 1.25, and 1.5 µg/mL, cell viability of BEAS-2B cells was reduced to 77%, 33%, 19%, 11%, and 5%, respectively; viability was significantly lower in the silver NP-exposed cells than in the control cells. The LD_50_ of 5 nm silver NPs was approximately 0.65 µg/mL. In contrast, 100 nm silver NPs at up to 1.5 µg/mL did not show cytotoxicity (Figure 2B). These results showed that silver NP toxicity was dependent on size and dose.

### 3.3. cDNA Microarray Analysis

Table 1 shows a summary of the differentially expressed gene count data 6 h post-silver NP exposure. The Venn diagram shows the number of genes with increased expression in EA.hy926 and BEAS-2B cells exposed to silver NPs at their approximate LD_50_ for 6 h (Figure 3A,B, respectively). Genes with >2-fold positive change in their absolute expression levels included oxidative stress- and cytokine production-related genes in 5 nm silver NP-treated cells (Table 2 and Table 3). Figure 4A,B show a heat map of the microarray results of IL-11, HSP-70, metallothionein 1G (MT1G), and HO-1 in EA.hy926 cells. Figure 4C,D show a heat map of the microarray results of IL-8, IL-11, HSP-70, MT1G, and HO (decycling)-1 expression changes in BEAS-2B cells.

### 3.4. Classification of Genes That Showed Increased Expression Following Treatment with Silver NPs

Genes that showed increased gene counts in EA.hy926 and BEAS-2B cells were divided into five categories based on the following gene ontology (GO) terms: cell death, cell survival, inflammation, apoptosis, and reactive oxygen species (ROS). IL-11 was included in the cell survival and inflammation categories. The results (Appendix A) suggested that 5 nm silver NPs activated inflammation and ROS stimulated expression of cell proliferation-related genes. Most of the stress-related genes were included in the cell death, apoptosis, and ROS categories.

### 3.5. Effects of Silver NPs on Cytokine Production and Inflammatory Response

Cells were treated with silver NPs at different concentrations for 8 h (Figure 5A,B) and 24 h (Figure 5C,D). IL-8 and IL-11 were expressed at 1000 pg/mL or more in EA.hy926 cells treated with 5 nm silver NPs at 2.5 µg/mL. IL-8 production was found to be the highest at 1413 pg/mL in cells treated with silver NPs at 0.5 µg/mL. IL-11 production was found to be highest at 3733 pg/mL in BEAS-2B cells treated with silver NPs at 0.75 µg/mL. As predicted, treatment with 100 nm silver NPs did not increase IL-8 or IL-11 release. Thus, cytokine expression was increased in cells in response to silver NPs in a size- and dose-dependent manner. Furthermore, these results supported the results of the cDNA microarray analysis.

### 3.6. Expression of Genes Related to Cytokine Production and ROS

At 5 nm silver NP concentrations of 1.5 and 2 µg/mL, IL-8 gene expression was increased by 3.5- and 6.5-fold, respectively, and IL-11 gene expression was increased by 2.9- and 6.2-fold, respectively, in EA.hy926 cells (Figure 6A). At the same doses, HSP-70 gene expression was increased by 3- and 27-fold, respectively, and HO-1 gene expression was increased by 9.8- and 19.7-fold, respectively (Figure 6B). Additionally, at 5 nm silver NP concentrations of 0.25, 0.5, and 0.75 µg/mL, IL-8 gene expression was increased by 2.8-, 20.6-, and 21.6-fold, respectively, and IL-11 gene expression was increased by 1.3-, 11.5-, and 15.6-fold, respectively, in BEAS-2B cells (Figure 6C). At the same doses, HSP-70 gene expression was increased by 16-, 85.6-, and 103.2-fold, respectively, and HO-1 gene expression was increased by 54.9-, 243.5-, and 163-fold, respectively, in BEAS-2B cells (Figure 6D).

### 3.7. Effects of Silver NPs on HSP-70 and HO-1 Expression

HSP-70 protein levels were increased by a maximum of 8.7-fold in EA.hy926 cells treated with 5 nm silver NPs at 2.75 µg/mL (Figure 7A), and HO-1 protein levels were increased by a maximum of 33.5-fold in EA.hy926 cells treated with 5 nm silver NPs at 3 µg/mL (Figure 7B), compared with in untreated EA.hy926 cells. HSP-70 protein levels were increased by a maximum of 7.7-fold and HO-1 protein levels were increased by a maximum of 8.9-fold in BEAS-2B cells treated with 5 nm silver NPs at 0.5 µg/mL, compared with in control cells (Figure 7C). In contrast, ROS-associated proteins were not affected by treatment with 100 nm silver NPs.

### 3.8. Intracellular Localization of Silver NPs

The transportation and localization of silver NPs in EA.hy926 cells were observed by TEM (Figure 8). A representative image of one EA.hy926 cell treated with silver NPs at 1.5 µg/mL for 30 min is shown. Untreated cells showed no abnormalities (Figure 8A), but 5 nm silver NP-treated cells showed internalization of NPs in vesicles (black arrow). Furthermore, mitochondrial swelling, decrease or disappearance of cytoplasm, and vacuolization were observed after treatment with 5 nm silver NPs (Figure 8B). After treatment with 100 nm silver NPs, no clear changes in cell morphology, including the presence of NPs in lysosomes and the nucleus, were observed (Figure 8C).

## 4. Discussion

In this study, the safety of silver NPs was investigated by examining the effects of silver NPs in endothelial and epithelial cells. Most previous studies anticipated that NP inhalation mainly affects the epithelial cells of respiratory tissues. However, a previous study verified that NPs introduced through the respiratory system were found in various other organs outside the respiratory system [7]. This indicated that NPs could penetrate epithelial cells to enter endothelial cells, owing to their extremely small size.

We evaluated the cytotoxic effects of silver NPs. Although the LD_50_ of NPs in each cell line was estimated to be different, the toxicity of silver NPs was confirmed. However, cell death was not observed in cells treated with 100 nm silver NPs at a high concentration (Figure 2). Therefore, it was suggested that the cytotoxicity of silver NPs depended on the particle size and treatment dose. We evaluated the effect of silver NPs in macrophage cells in our previous study. We verified that 5 nm silver NPs affected the variation in gene expression in cells through microarray analysis and observed a noticeable increase in IL-8 and oxidative-stress related gene expression in early stages [19]. Similarly, we verified that oxidative stress- and inflammation-related gene expression was increased in 5 nm silver NP-exposed cells. ROS are induced by oxidative stress during cell death or in early stages before cellular damage [20]. We also observed noticeable variation in the expression of oxidative stress-related genes, especially MT, HO-1, and HSP-70, in early stages after exposure. In a previous study, 20–50 nm silver NPs were added to A549 cells, thereby producing oxidative stress products, such as malondialdehyde and 8-hydroxy-2′-deoxyguanosine, and consequently increasing HSP-70 and HO-1 levels [21]. These results were similar to those of our study.

The microarray used in this study can analyze up to 24,838 RefSeq (Entrez) genes. Based on the results of this analysis, genes undergoing intracellular change after exposure to silver NPs were classified into five categories: cell death, cell survival, inflammation, apoptosis, and ROS. As results, there were considerable variations in genes related to cell death and apoptosis as well as genes related to cell survival. Meanwhile, the expression of genes related to inflammation was relatively low. This is supported by our review of previous research, which showed that only IL-8 gene expression among cytokines increased significantly after the exposure of macrophage cells to 5 nm silver NPs [20]. In epithelial cells, the only intracellular cytokines that increased following exposure to 5 nm of silver NPs were IL-8 and IL-11. Additionally, ROS-related genes were most marked as ion-binding GO terms, probably because silver NPs are gradually ionized to Ag+ over time and variation in ion-binding-related gene expression develops actively. A previous study suggested that toxicity was generated by free silver ions induced by silver NPs, demonstrating the combined effect of silver ions and silver NPs [22].

Notably, the increased IL-11 gene expression observed in this study has not been greatly discussed in previous studies. In this study, IL-11 gene expression was particularly increased in both endothelial and epithelial cells following exposure to 5 nm silver NPs. As an anti-inflammatory cytokine, IL-11 has been identified to exert immunomodulatory activities by reducing pro-inflammatory cytokine synthesis. Owing to its pivotal role in inflammatory conditions, IL-11 has been investigated as a therapeutic candidate for rheumatoid arthritis, Crohn’s disease, refractory immune thrombocytopenic purpura, and periodontal disease [23]. Thus, IL-11 has various functions and plays various roles in different cell types [18]. One study suggested that IL-11 was involved in endothelial cell proliferation, indicating that IL-11 was associated with the STAT3 pathway, and accordingly, survival proteins were expressed and exerted positive effects on cell proliferation [24]. However, the microarray results in this study showed increased expression of 31 genes, including IL-11, in epithelial cells and 20 genes, including IL-11, in endothelial cells, and these genes were classified as the regulation of cell proliferation GO term. It was inferred that IL-11 played a role in cell proliferation of not only endothelial cells, but also epithelial cells. Moreover, IL-11 is classified as an anti-inflammatory cytokine; the microarray results for endothelial cells showed a 10.4-fold increase in IL-11 expression, and this expression was much higher than that of other cytokines. Additionally, genes classified as the regulating apoptosis GO term showed increased expression by 37-fold in endothelial cells and 49-fold in epithelial cells. Although IL-11, which is associated with apoptosis, induced cell death in both endothelial and epithelial cells, it stimulated more genes in epithelial cells. This finding is supported by results of earlier studies, which showed that one of the various functions of IL-11 was the induction of apoptosis of epithelial cells [12,25,26]. Given the various roles played by IL-11, it is difficult to determine its role in specific cells. Therefore, although this study verified that IL-11 expression was commonly increased in 5 nm silver NP-exposed cells, additional studies using various methods are required in the future to determine the specific roles played by IL-11 based on the cell type.

## 5. Conclusions

This study showed that intracellular genes specifically responded to silver NPs, and the expression of IL-11 among all cytokines, exclusive of IL-8, was significantly increased.

## Figures and Tables

**Figure 1 biomolecules-11-00234-f001:**
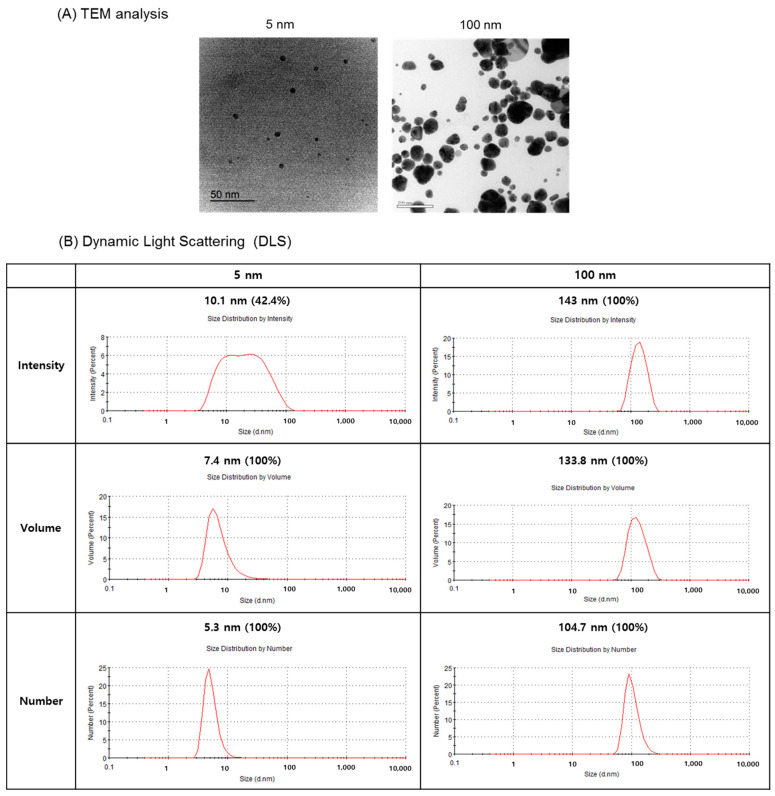
Analysis of silver nanoparticles (NPs). (**A**) Transmission electron microscopy images show that the 5 nm silver NPs were relatively uniform in size, whereas the 100 nm silver NPs were varied in size. (**B**) For DLS analysis, silver NPs were dispersed in DMEM containing 10% FBS.

**Figure 2 biomolecules-11-00234-f002:**
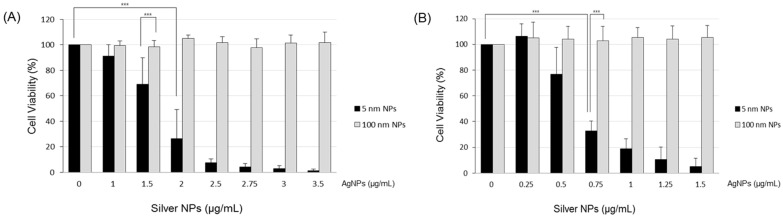
Cytotoxicity of silver NPs. The median lethal dose of 5 nm silver NPs was approximately (**A**) 1.72 µg/mL in EA.hy926 cells and (**B**) 0.65 µg/mL in BEAS-2B cells. Data represent the mean ± standard deviation of three independent experiments. One-way and two-way analysis of variance was performed to determine the significance of the differences (*** *p* < 0.001).

**Figure 3 biomolecules-11-00234-f003:**
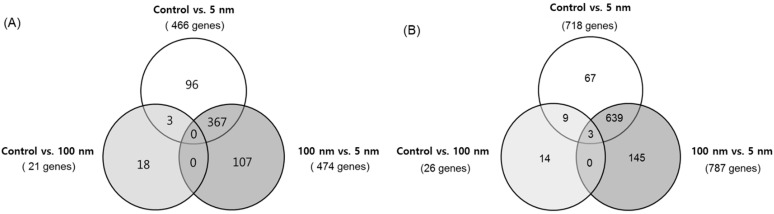
Microarray analysis to determine increased gene expression changes. (**A**) EA.hy926 cells were exposed to silver NPs at their approximate LD_50_ (2 µg/mL) for 6 h. (**B**) BEAS-2B cells were exposed to silver NPs (0.5 µg/mL) for 6 h. Results were analyzed using a *t*-test, and significance was set at *p* < 0.05.

**Figure 4 biomolecules-11-00234-f004:**
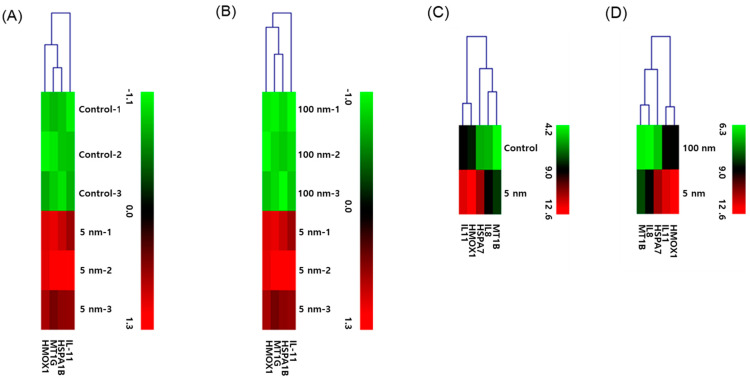
Hierarchical clustering. Microarray analysis of (**A**) untreated EA.hy926 cells vs. 5 nm NP-treated EA.hy926 cells, (**B**) 100 nm NP-treated EA.hy926 cells vs. 5 nm NP-treated EA.hy926 cells, (**C**) control vs. 5 nm NP-treated BEAS-2B cells, and (**D**) 100 nm NP-treated BEAS-2B cells vs. 5 nm NP-treated BEAS-2B cells is presented. Results were analyzed using a t-test, and significance was set at *p* < 0.05.

**Figure 5 biomolecules-11-00234-f005:**
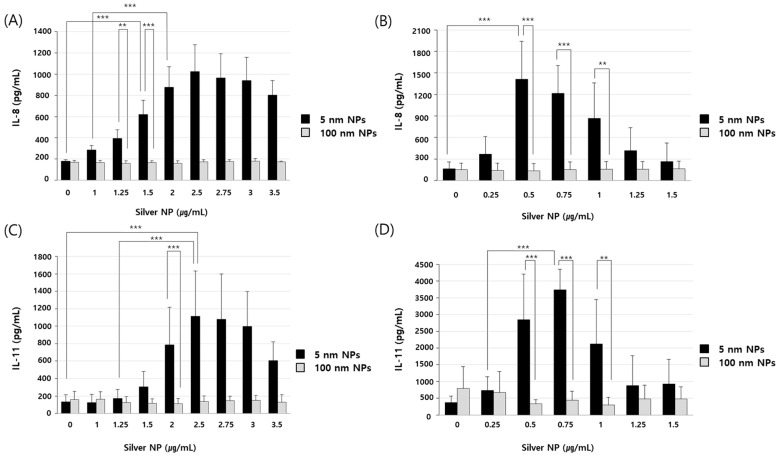
Dose-dependent expression of cytokines in response to silver NPs. (**A**) EA.hy926 and (**B**) BEAS-2B cells were treated with silver NPs at different doses for 8 h. (**C**) EA.hy926 and (**D**) BEAS-2B cells were treated with silver NPs at different doses for 24 h. Data represent the mean ± standard deviation of three independent experiments. One-way and two-way analysis of variance was performed to determine the significance of the differences (** *p* < 0.01, *** *p* < 0.001).

**Figure 6 biomolecules-11-00234-f006:**
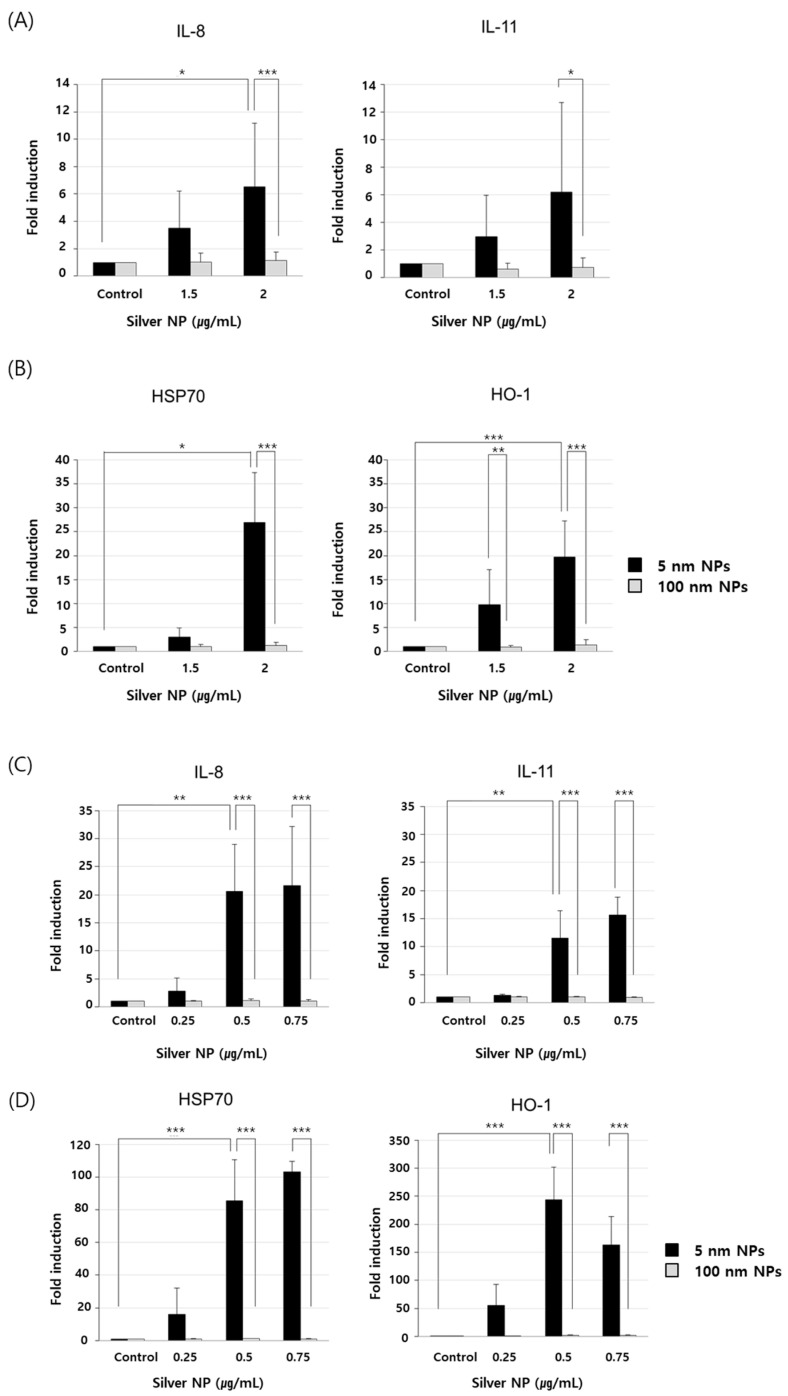
Expression of genes related to cytokine production and reactive oxygen species (ROS). (**A**,**B**) RT-PCR analysis was performed for EA.hy926 cells treated with silver NPs at 1.5 and 2 µg/mL for 6 h. (**C**,**D**) Real-time RT-PCR analysis was performed for BEAS-2B cells treated with silver NPs at 0.25, 0.5, and 0.75 µg/mL for 6 h. Data are shown as the mean ± standard deviation of three or more independent experiments. One-way and two-way analysis of variance was performed to determine the significance of the differences (* *p* < 0.05, ** *p* < 0.01, *** *p* < 0.001).

**Figure 7 biomolecules-11-00234-f007:**
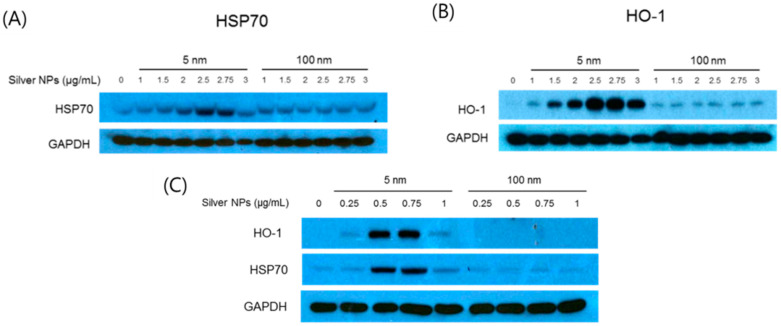
Heat shock protein 70 kDa (HSP-70) and hemeoxygenase (HO)-1 protein levels in silver NP-treated human cells. (**A**,**B**) EA.hy926 cells were treated with silver NPs at different doses for 24 h. Each 40-µg protein sample was loaded and analyzed using anti-HSP-70, anti-HO-1, and anti-glyceraldehyde-3-phosphate dehydrogenase (GAPDH) (loading control) antibodies. (**C**) BEAS-2B cells were treated with silver NPs at different doses for 24 h. Each 30-µg protein sample was loaded and analyzed using anti-HSP-70, anti-HO-1, and anti-GAPDH (loading control) antibodies.

**Figure 8 biomolecules-11-00234-f008:**
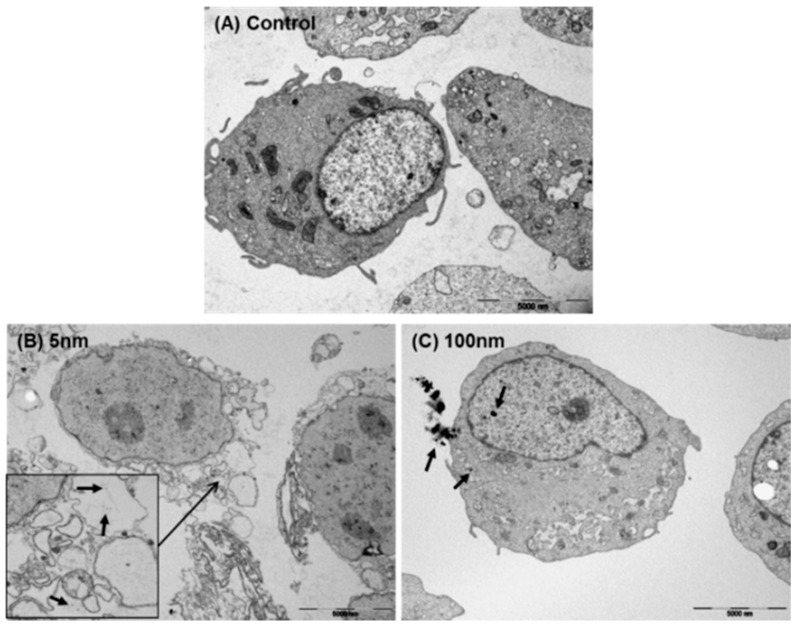
Transmission electron microscopy images of intracellular silver NP localization. (**A**) Untreated cells showed no abnormalities. (**B**) Cells treated with 5 nm silver NPs showed NP internalization, mitochondrial swelling, decrease or disappearance of cytoplasm, and vacuolization. (**C**) Cells treated with 100 nm silver NPs showed no clear changes in cell morphology. Images shown are representative of three independent trials.

**Table 1 biomolecules-11-00234-t001:** Summary of differentially expressed genes.

Group	EA. hy926 ^1^			BEAS-2B ^2^		
	Up-Regulation	Down-Regulation	Total Genes	Up-Regulation	Down-Regulation	Total Genes
Control vs. 5 nm	466	193	659	726	242	968
Control vs. 100 nm	27	20	47	39	55	94
100 nm vs. 5 nm	479	181	660	821	251	1072

^1^ Significant transcripts with expression values changed by 1.5-fold or greater were selected. ^2^ Significant transcripts with expression values changed by 2.0-fold or greater were selected. A t-test was performed to evaluate the significance of the differences.

**Table 2 biomolecules-11-00234-t002:** List of increased genes after silver NP exposure in EA.hy926 cells.

**(A) Control vs. 5 nm**		
**Genes**	**Fold Change**	***p*-Value**
metallothionein 1G	47.9	0.0008
long intergenic non-protein coding RNA 622	15.1	0.0079
heme oxygenase (decycling) 1	13.3	0.0001
putative novel transcript	11.3	0.0189
interleukin 11	10.4	0.0015
matrix metallopeptidase 10 (stromelysin 2)	8.7	0.0040
chemokine (C-X-C motif) ligand 8	6.1	0.0045
heat shock 70 kDa protein 1B	5.1	0.0004
heat shock 70 kDa protein 6 (HSP70B)	4.8	0.0027
metallothionein 1E	4.6	0.0008
interleukin 36, alpha	4.6	0.0292
heat shock 70 kDa protein 1A	4.0	0.0006
interleukin 13 receptor, alpha 2	3.8	0.0474
heat shock 70 kDa protein 9 (mortalin)	3.5	0.0082
interleukin 1 receptor-like 1	3.1	0.0080
interleukin 7 receptor	2.7	0.0143
interleukin 1, alpha	2.6	0.0215
vascular endothelial growth factor A	2.4	0.0283
metallothionein 1X	2.3	0.0019
metallothionein 1F	2.2	0.0379
**(B) 100 nm vs. 5 nm**		
**Genes**	**Fold Change**	***p*-value**
metallothionein 1G	44.3	0.0008
long intergenic non-protein coding RNA 622	18.1	0.0061
putative novel transcript	13.8	0.0140
heme oxygenase (decycling) 1	11.5	0.0001
interleukin 11	11.0	0.0013
nuclear receptor subfamily 4, group A, member 1	7.9	0.0003
matrix metallopeptidase 10 (stromelysin 2)	7.8	0.0028
chemokine (C-X-C motif) ligand 8	6.4	0.0008
heat shock 70 kDa protein 6 (HSP70B)	5.2	0.0026
heat shock 70 kDa protein 1B	5.0	0.0005
metallothionein 1E	4.3	0.0008
interleukin 36, alpha	4.0	0.0391
heat shock 70 kDa protein 9 (mortalin)	3.9	0.0042
heat shock 70 kDa protein 1A	3.9	0.0008
interleukin 13 receptor, alpha 2	3.5	0.0479
interleukin 1 receptor-like 1	2.8	0.0108
spermine oxidase	2.8	0.0069
vascular endothelial growth factor A	2.6	0.0228
interleukin 1, alpha	2.6	0.0191
metallothionein 1X	2.3	0.0010

Significant transcripts were selected when expression values changed by 2-fold or greater and while using a *t*-test with *p*-value < 0.05.

**Table 3 biomolecules-11-00234-t003:** List of increased genes after silver NP exposure in BEAS-2B cells.

**(A) Control vs. 5 nm**	
**Genes**	**Fold Change**
activity-regulated cytoskeleton-associated protein	46.1
heat shock 70 kDa protein 7 (HSP70B)	36.4
heme oxygenase (decycling) 1	18.1
metallothionein 1B	12.8
metallothionein 1E	12.7
heat shock 70 kDa protein 6 (HSP70B’)	11.1
interleukin 8	9.8
metallothionein 1G	6.8
interleukin 11	6.2
interleukin 1 receptor-like 1	5.4
metallothionein 1M	5.1
heat shock 70 kDa protein 1B	4.2
BCL2-associated athanogene 3	3.7
heat shock 70 kDa protein 1A	3.5
cytochrome P450, family 4, subfamily F, polypeptide 11	3.4
heat shock 70 kDa protein 9 (mortalin)	3.4
VGF nerve growth factor inducible	3.1
oxidative stress responsive serine-rich 1	2.7
heparin-binding EGF-like growth factor	2.6
Smad nuclear interacting protein 1	2.4
**(B) 100 nm vs. 5 nm**	
**Genes**	**Fold Change**
activity-regulated cytoskeleton-associated protein	48.0
heat shock 70 kDa protein 7 (HSP70B)	28.9
heme oxygenase (decycling) 1	12.6
interleukin 8	11.5
heat shock 70 kDa protein 6 (HSP70B’)	9.9
metallothionein 1G	6.0
interleukin 11	6.0
metallothionein 1B	5.4
interleukin 1 receptor-like 1	5.4
metallothionein 1E	4.7
heat shock 70 kDa protein 1-like	4.5
heat shock 70 kDa protein 1B	4.4
metallothionein 1M	3.8
heat shock 70 kDa protein 9 (mortalin)	3.7
BCL2-associated athanogene 3	3.6
heat shock 70 kDa protein 1A	3.4
VGF nerve growth factor inducible	3.1
BCL2-related protein A1	2.4
interleukin 6 receptor	2.2
heat shock 22 kDa protein 8	2.1

Significant transcripts were selected when expression values changed by 2-fold or greater.

## Data Availability

The data presented in this study are openly available in this article.

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
