# Peer review of "Increased Interleukin-11 and Stress-Related Gene Expression in Human Endothelial and Bronchial Epithelial Cells Exposed to Silver Nanoparticles"

_biomolecules, 2021, doi:10.3390/biom11020234_

Round 1

Reviewer 1 Report

1. In the Introduction there should be some mention about why other parameters than interleukin-11 expression have been chosen in this work, as in the manuscript body it is not discussed in much more detail than other parameters (e.g. IL-8 and expression of genes connected with oxidative stress). So, it is not clear why this one cytokine has been singled out.
2. In the Experimental part (chapter 2.9) the concentration of NPs for micrograms/mL cells is given as a range of 1.5-2 micrograms/mL. However, from the discussion it seems that there were only these two concentration values, so it would be more unequivocal if it would be put as 1.5 and 2 micrograms/mL. For BEAS-2B cells in chapter 2.9 there is only one concentration given, while in the Results section, in Fig. 6 there are three concentrations shown.
3. In most figures the font is too small, and it is difficult to analyze the results. The graphical ilustration of DLS results in Fig. 1B is of too low quality to see anything.
4. Please add what was the size distribution in 100 nm NPs.
5. In chapter 3.3 please add a justification why these specific genes have been chosen for heat maps.
6. In the caption of Fig. 3, which shows the Venn diagram, there is a sentence: "Gene expression changes are presented on an absolute fold change scale, where each unit represents a 1.5-fold change in expression level". However, in this type of diagram no quantitative results are shown, and I can't see any units here. Please remove this sentence or show the units.
7. In the Results section (chapters 3.6 and 3.7) and in Discussion please specify where appropriate the size of nanoparticles that caused the described effects, as in the present form it gives the impression that for both 5 nm and 100 nm NPs these results are valid - and it's not consistent with data in Figures 6 and 7.
8. Please discuss shortly also the IL-8 expression changes.
9. In Conclusion it is stated that: "the expression of IL-11, among all cytokines, was significantly increased" - it suggests that the increase of IL-11 expression was significantly higher than that of other cytokines in both studied systems, which is not consistent with results presented in Table 3 and Fig.6.

Reviewer 2 Report

The “Increased interleukin-11 and stress-related gene expression in human endothelial and bronchial epithelial cells exposed to silver nanoparticles “manuscript is an investigation of genomic change of EA.hy926 cells and BEAS-2B cells lines exposed to 2 different size of silver nanoparticles, in an attempt to explain the mechanism of silver toxicity. The toxicity of particles was assessed up to 3 ug/ml concentration and gene expression changes were evaluated by PCR, ELISA, and western blotting. The manuscript is well organized and well written. However, the nanoparticles were not characterized fully, and 100nm silver nanoparticle bought from ABC Nanotech company has high polydispersity.

  • Please explain the rational of timing (6h) and dosing (LD50) for gene expression study.

  • Gene expression were studied at LD50 for 5 nm particles, however, at same dose, the 100nm is not toxic in either cell lines. Please explain what is the rational for comparing the toxic dose of 5nm silver nanoparticles with none toxic 100nm.

  • Please elaborate effect of the LD50 of 100nm silver nanoparticle by increasing the treatment does of nanoparticles and difference of mechanism of toxicity of 100nm compared to 5nm nanpoparticle.

  • Validating experiments such as cytokine production were studied at time points (8h instead of 6h) and concentration (2.5ug/ml instead of 1.75ug/ml) different than the gene expression study. To obtain a better picture of mechanism of toxicity it would have been better to validate the gene expression studies at same time points and concentration.
